

# Exploring the observational constraints on the simulation of brown carbon

Xuan Wang[1], Colette L. Heald[1, 2], Jiumeng Liu[3], Rodney J. Weber[4], Pedro Campuzano-Jost[5,6], Jose L. Jimenez[5,6], Joshua P. Schwarz[7], and Anne E. Perring[7]

[1]Department of Civil and Environmental Engineering, Massachusetts Institute of Technology, Cambridge, MA, USA
[2]Department of Earth, Atmospheric and Planetary Sciences, Massachusetts Institute of Technology, Cambridge, MA, USA
[3]Atmospheric Sciences and Global Change Division, Pacific Northwest National Laboratory, Richland, WA, USA
[4]School of Earth and Atmospheric Sciences, Georgia Institute of Technology, Atlanta, GA, USA
[5]Department of Chemistry, University of Colorado Boulder, Boulder, CO, USA
[6]Cooperative Institute for Research in Environmental Sciences, University of Colorado Boulder, Boulder, CO, USA
[7]Chemical Sciences Division, Earth System Research Laboratory, National Oceanic and Atmospheric Administration, Boulder, CO, USA

*Correspondence to*: Xuan Wang (xuanw12@mit.edu)

**Abstract.** Organic aerosols (OA) that strongly absorbs solar radiation in the near-UV are referred to as brown carbon (BrC). However the sources, evolution, and optical properties of BrC remain highly uncertain, and contribute significantly to uncertainty in the estimate of the global direct radiative effect (DRE) of aerosols. Previous modeling studies of BrC optical properties and DRE have been unable to fully evaluate model performance due to the lack of direct measurements of BrC absorption. In this study, we develop a global model simulation (GEOS-Chem) of BrC and test it against BrC absorption measurements from two aircraft campaigns in the continental U.S. (SEAC4RS and DC3). To our knowledge, this is the first study to compare simulated BrC absorption with direct ambient measurements. We show that the BrC absorption properties from biomass burning estimated based on previous laboratory measurements overestimate the aircraft measurements of ambient BrC absorption. In addition, applying a photochemical scheme to simulate bleaching/degradation of BrC improves model skill. The airborne observations are consistent with a mass absorption coefficient (MAC) of freshly emitted biomass burning OA of $0.57 m^2 g^{-1}$ at 365nm with an absorption Ångström exponents (AAE) = 3.1 for 300&600 wavelengths pair, coupled with a 1 day whitening e-folding time. Using the GEOS-Chem chemical transport model integrated with the RRTMG radiative transfer model, we estimate that the top-of-atmosphere all-sky direct radiative effect (DRE) of OA is -$0.350$ Wm$^{-2}$, 10% higher than that without consideration of BrC absorption. Therefore, our best estimate of the absorption DRE of BrC is +$0.042$ Wm$^{-2}$. We suggest that the DRE of BrC has been overestimated previously due to the lack of observational constraints from direct measurements and omission of the effects of photochemical whitening.

## 1 Introduction

Carbonaceous aerosols, including both black carbon (BC) and organic aerosols (OA), are among the largest sources of uncertainty in the estimate of the global direct radiative effect (DRE) and forcing (DRF) of aerosols. BC is the principle



light-absorbing aerosol in the atmosphere, whereas OA is generally considered as "white carbon" which scatters light without corresponding absorption. However, a fraction of OA is also found to efficiently absorb light, predominately at near-UV wavelengths (Kirchstetter et al., 2004; Hecobian et al., 2010; Arola et al., 2011). This absorbing OA, referred to as brown carbon (BrC), has primarily been associated with biofuel or biomass combustions (Andrea et al., 2006; Ramanathan et

al., 2007; Washenfelder et al., 2015). These BrC emissions are typically mixed with co-emitted BC and non-absorbing OA, challenging the measurement community's ability to evaluate the optical properties of ambient BrC. Additional sources of BrC, including the photo-oxidation of volatile organic compounds (VOCs) and aqueous-phase chemistry in cloud droplets, typically produce less absorbing BrC, with properties that are even more uncertain (Graber et al., 2006; Ervens et al., 2011; Wang et al., 2014; Laskin et al., 2015). A few studies have attempted to simulate BrC in global models and estimate its DRE

(Park et al., 2010, Feng et al., 2013; Lin et al., 2014; Wang et al., 2014; Jo et al., 2015; Saleh et al., 2015). These estimates range from +0.1 to +0.6 $Wm^{-2}$, corresponding to 20% to 40% of the total absorption of carbonaceous aerosol across studies. Due to our poor understanding of the sources, optical properties, chemistry, and mixing state of BrC, the uncertainty surrounding the global absorption from BrC remains high.

Most modelling studies follow a similar approach to simulating BrC: some fraction of OA is assumed to be BrC, and

assigned different optical properties from non-absorbing OA. The assumed optical properties of BrC are based on laboratory measurement of organics extracted in water, acetone, methanol, or other organic solvents. However, these properties are not well constrained by laboratory studies.  First, measured absorption properties differ significantly among studies (Wang et al., 2014). Even within a study, different combustion conditions (e.g. burning temperature) can also lead to up to a factor of two difference in absorption properties (e.g. the imaginary part of refractive index, or mass absorption coefficient) (Chen and

Bond, 2010). Previous modelling studies have typically used either the lower or higher bound from laboratory studies to estimate the minimum or maximum absorption properties of BrC (Feng et al., 2013; Lin et al., 2014). In addition, it is unclear what fraction of the OA is BrC and how this differs with source and ambient combustion conditions (Pokhrel et al., 2017). In laboratory studies organics are not always fully soluble; typically, 40%-90% of the total material can be extracted, depending on the solvent (e.g. ~40% can be extracted in water and more than 90% can be extracted in methanol, Chen and

Bond, 2010). The absorption properties of the insoluble fraction are unknown. Thus previous modelling studies have applied a range of assumptions: Lin et al., (2014) assumed all primary organic aerosols (POA) from biofuel and biomass emissions and all secondary organic aerosols (SOA) from biogenic and anthropogenic emissions to be BrC; Feng et al. (2013) assumed 66% of biofuel/biomass POA to be BrC; Wang et al. (2014) assumed 25% of biomass burning and 50% of biofuel POA as well as aromatic SOA were brown. There is tenuous scientific support for these assumptions. In addition, extrapolating

laboratory experiments to real-world combustion sources may also lead to large uncertainties.

Recent studies show that the BrC absorption from biofuel or biomass sources is likely affected by combustion efficiency (Chen and Bond, 2010; Saleh et al., 2014; Pokhrel et al., 2016; 2017). A number of modelling studies have attempted to connect BrC absorption to emission properties, by using the modified combustion efficiency (MCE, a function of $CO/CO_2$) (Jo et al., 2016), or the BC/OA ratio (Park et al., 2010; Saleh et al., 2015). These approaches should better represent the



temporal and spatial variability of BrC emissions and properties, however, in practice these parameterizations are difficult to apply in models given the lack of information regarding burn conditions in emissions inventories. The variability of quantities such as BC/OA or MCE in these inventories reflects differences in fuel types (and the associated emission factors), not burn conditions. Therefore, these studies fail to describe the variation in emissions of BrC within a given fuel

type.

Both of these methods focus primarily on the sources of BrC, however chemical transformation and the mixing state of BrC also play an important role in controlling BrC absorption. In laboratory studies, the absorption of BrC is found to both increase during the formation or chemical aging of certain types of OA, and decrease during oxidation or photolysis (Zhong and Jiang, 2011; Flores et al., 2014; Lee et al., 2014; Liu et al., 2016). Field studies provide evidence that BrC may be

formed in clouds or during convective transport, due to aqueous phase chemistry or condensation (Gilardoni et al., 2016; Zhang et al., 2017).  Observations also indicate that biomass burning BrC absorption decreases with photochemical aging with a lifetime of ~1 day (Forrister et al., 2015; Wang et al., 2016). It is likely that the absorption and DRE of BrC would change significantly if these chemical processes were included in models. For the mixing state, the key question is whether BrC is internally or externally mixed with BC. Previous studies typically assume that BrC is externally mixed with BC (Liu

et al., 2013). When considering BC only, the internal mixing is widely idealized as acore-shell morphology (Jacobson, 2001; Bond and Bergstrom, 2006). When coated by other materials, typically inorganic and non-absorbing OA, the absorption of BC will be enhanced by the lensing effect (Jacobson, 2001; Bond et al., 2006). However, if BrC coats BC, Mie calculations show a lower absorption enhancement for BC. At the same time, the absorption of BrC itself will decrease since there is less externally-mixed BrC left in the atmosphere. As a result, the mixing state of BrC will affect the absorption of not only the

BrC but also the BC. This influence is sensitive to the absorption properties of BrC, which are highly uncertain, as we have discussed, and also the proportion of externally/internally mixed BrC, which to our knowledge, has not been measured in the atmosphere. Saleh et al. (2015) investigate this influence and conclude that for a single particle with fixed size (BC=150nm and BrC=200nm) the global mean absorption DRE of BrC decreases by 45% when assuming complete internal mixing (compared to complete external mixing).

Observational constraints on BrC are scarce, thus making it a challenge to test and improve models based on observational evidence. Although the absorption of aerosols is widely measured in the form of absorption aerosol optical depth (AAOD) by satellite or ground-based measurement, these observations include the absorption of both BrC and other aerosols (primarily BC). In our previous work (Wang et al., 2016), we presented a method to distinguish the absorption contributions of BrC and BC. However, the method can only be used for multiple-wavelength absorption observations with 2 wavelengths

longer than 600nm and at least 1 in the near-UV. Such measurements are currently limited and exhibit large uncertainties. In addition, absorption measurement at very low wavelengths where BrC dominates absorption would also help constrain the abundance and properties of BrC, however these wavelengths are not available for current remote sensing observations. Recently, during two aircraft campaigns (DC3 and SEAC4RS, see details in Section 2) BrC absorption was directly measured. This provides an opportunity to test the model assumptions. However, properties of BrC, including absorption,





chemical transformation, and mixing state, are still challenging to evaluate because of the uncertainty surrounding the simulation of OA mass. Models fail to reproduce the observed magnitude and variation of OA mass concentrations (Heald et al., 2011; Spracklen et al., 2011; Tsigaridis et al., 2014). Thus, it can be challenging to untangle whether any discrepancy between modelled and observed BrC absorption should be attributed to BrC properties or OA mass concentrations.

Furthermore uncertainties surrounding the simulation of BC (Koch et al., 2009; Bond et al., 2013; Wang et al., 2014), may also impact a combustion-based approach (MCE or BC/OA) to simulating BrC.

Given the above context, it is highly challenging to develop and test an accurate model simulation of BrC. A reasonable approach is to test the simplest assumptions for BrC modelling. In this study, we develop a model simulation of BrC, test it against BrC absorption measurements from two aircraft campaigns in the U.S. (SEAC4RS and DC3), and optimize it to

match these observational constraints. To our knowledge, this is the first study to compare simulated BrC absorption and its vertical variation with direct, continuous aircraft measurements. We explore how assumptions for BrC sources, processing, and properties impacts the comparisons with these observational constraints and estimate the resulting global direct radiative effect of BrC under these conditions.

## 2 Aircraft observations

In this study, we compare our model results to the DC-8 airborne measurements from two campaigns: DC3 and SEAC4RS. The DC3 (Deep Convective Clouds and Chemistry) campaign was conducted from May 18$^{th}$ to June 22$^{nd}$ in 2012, over the central and southeastern U.S (Barth et al., 2015). The SEAC4RS (Studies of Emissions, Atmospheric Composition, Clouds and Climate Coupling by Regional Surveys) campaign occurred in a similar region during August 6$^{th}$ to September 23$^{rd}$ in 2013 (Toon et al., 2016). Flight tracks are shown in Figure 1. BrC absorption and related aerosol measurements of interest

were made by the same instruments during these two campaigns.

The OA absorption (hence the BrC) was directly measured from liquid extracts of aerosol samples. The samples with aerodynamic diameter smaller than 4.1μm were collected on Teflon filters every 5 minutes. Water extracts were transferred to an LWCC-TOC system (Liquid Waveguide Capillary Cell coupled to a Total Organic Carbon analyzer). The absorption spectra of the extracts were measured in the 200 to 800nm wavelength range; these measurements are referred to as

H$_2$O_Abs. The detection limit and uncertainty of H$_2$O_Abs is 0.031 Mm$^{-1}$ and 20% respectively, at 365nm. The insoluble fraction of the samples was sequentially extracted in methanol following the same method as water extracts. This part of the absorption is referred to as MeOH_Abs and has a detection limit and uncertainty of 0.11 Mm$^{-1}$ and 37% at 365nm. The total absorption of OA is determined by summing H$_2$O_Abs and MeOH_Abs as reported at 365nm. A multiplication factor of 2 is applied here to convert the solution absorption to aerosol absorption, reflecting the enhanced absorption by aerosols in the

Mie regime versus molecules in the liquid extract (Zhang et al., 2017). An important assumption here is that essentially all of the BrC can be extracted in water and methanol, which is supported by laboratory experiments (Chen and Bond, 2010). Further details on these measurements can be found in Liu et al. (2015).





In addition to OA absorption, the mass concentrations of aerosols and gases were measured throughout the two campaigns. Sub-micron OA (and inorganic aerosols) were measured by a high resolution time-of-flight Aerodyne Aerosol Mass Spectrometer (AMS, DeCarlo et al., 2006) with an estimated uncertainty of 38%. The transmission of particles through the AMS aerodynamic lens is ~100% on the range 50-550 nm and then declines up to above 1 μm, and is referred to approximately as $PM_1$ (Dunlea et al., 2009). BC accumulation mode mass concentrations were measured with a Single Particle Soot Photometer (SP2, Schwarz et al., 2008) with an estimated uncertainty of 30%; these measurements were made off a different inlet and sampling line with good transport efficiency only up to 3 μm total particle diameter (50% transport efficiency at 3 μm). Carbon monoxide (CO) and acetonitrile ($CH_3CN$) were measured with a Diode laser spectrometer and PTR-MS with uncertainties of 2% and 20%, respectively. Details on all of these measurements, as well as other aerosol and gas measurements made during the campaigns, can be found in Barth et al. (2015) and Toon et al. (2016).

## 3 Model description

### 3.1 The GEOS-Chem model with RRTMG

We use the global chemical transport model GEOS-Chem (Bey et al., 2001) coupled with the rapid radiative transfer model for GCMs (RRTMG, Iacono et al., 2008) in this study. Our simulations use the GEOS-FP assimilated meteorology from the Goddard Earth Observing System (GEOS) at the NASA Global Modeling and Assimilation office. The global simulations use v10-1 of GEOS-Chem with a horizontal resolution of 2 °×2.5 ° and 47 vertical levels. When comparing with aircraft measurements, we perform nested simulations over North America (10–60 °N, 130–60 °W) at 0.25 °×0.3125 ° horizontal resolution. RRTMG is a radiative transfer model which calculates both longwave and shortwave atmospheric radiative fluxes. This calculation is coupled to GEOS-Chem and conducted every 3 hours. Details of the implementation of RRTMG in GEOS-Chem are available in Heald et al. (2014).

The simulation of POA and BC mass are based on the standard GEOS-Chem simulation with modifications described in Wang et al. (2014). The model assumes 50% of anthropogenic and 30% of emitted biomass burning organic carbon (OC) is hydrophobic and the remaining is hydrophilic. Hydrophobic OC converts to hydrophilic OC with an e-folding time of 1.15 days, equal to an aging rate of ~$10^{-5}$ s$^{-1}$. The POA is inferred from simulated primary OC by applying an OA/OC mass ratio of 2.1 (Turpin and Lim, 2001; Aiken et al., 2008; Canagaratna et al., 2015). This represents average atmospheric OA/OC composition. Freshly emitted POA is less oxidized (1.34-1.65; Canagaratna et al., 2015), however aging occurs quickly in the atmosphere, in particular for biomass burning OA (Cubison et al., 2011; Forrister et al., 2015). The simulation of BC includes a source-specific treatment. For the fossil BC, we assume 80% are emitted as hydrophobic, and convert to hydrophilic with an aging rate related to $SO_2$ and OH levels in the atmosphere:

$$k = \propto [SO_2][OH] + b \qquad (1)$$

where $\alpha = 2 \times 10^{-22}$cm$^6$molec$^{-2}$s$^{-1}$ and b = 5.8 $\times 10^{-7}$s$^{-1}$ (Liu et al., 2011; Wang et al., 2014). The biofuel/biomass burning BC is assumed to be emitted as 70% hydrophilic and 30% hydrophobic with an aging e-folding time from hydrophobic to



hydrophilic of 4 hours. The details of the BC scheme and evaluation against BC mass concentrations can be found in Wang et al. (2014). Our simulation of SOA is from the standard GEOS-Chem simulation, which is based on reversible partitioning of semivolatile products of aromatic and biogenic VOC oxidation (Pye and Seinfield, 2010; Pye et al.,2010).

The global anthropogenic emissions of BC and POA follow the Bond et al. (2007) emission inventory (8.7 TgC yr$^{-1}$ for POA

and 4.4 TgC yr$^{-1}$ for BC, globally). For the North America region, The EPA National Emission Inventory for 2011 (EPA/NEI11) is used. We also implement the annual scaling factors from the EPA's air pollutant emissions trends data (https://www.epa.gov/air-emissions-inventories/air-pollutant-emissions-trends-data) and a 17% decrease for SO$_2$ and a 30% decrease for BC, suggested by Kim et al. (2015), who conducted an analysis of the aerosols during SEAC4RS. The resulting anthropogenic POA and BC emissions from the contiguous United States total 0.58 TgC yr$^{-1}$ and 0.26 TgC yr$^{-1}$ for 2013.

Since the EPA/NEI11 inventory does not separate fossil and biofuel emissions, we apply the fossil/biofuel emission ratios from the Bond et al. (2007) emission inventory and the seasonal cycle of residential emission from Park et al. (2003) to separate these two emission types. This produces an annually averaged fossil/biofuel emission ratio of 8 for BC and 1.2 for OC in US. The biomass burning emissions of BC and POA follow the year-specific daily mean GFED4s (Global Fire Emissions Database with small fires) inventory (Van der Werf et al., 2010, Giglio et al., 2013), contributing 0.1 TgC yr$^{-1}$ of

BC and 1.42 TgC yr$^{-1}$ of POA in US in 2012, 0.15 TgC yr$^{-1}$ of BC and 2.24 TgC yr$^{-1}$ of POA in US in 2013. The biogenic VOC emissions are simulated online based on the MEGAN2.1 (Model of Emissions of Gases and Aerosols from Nature) scheme (Guenther et al., 2012). The anthropogenic VOC emissions are based on the combination of Reanalysis of the Tropospheric chemical composition (RETRO) global emission inventory (Pulles et al., 2007) and EPA/NEI11 inventory for United States.

**3.2 Treatment of BrC optical properties**

Most previous BrC modelling studies assume some fraction of OA to be BrC and assign it different optical properties from non-absorbing OA. Unlike this approach, we assign absorption properties for all OA, thus convolving two unknowns into a single assumption. This simplifies our analysis, given that the total absorption from OA equals the absorption from BrC:

$$Abs_{BrC} = Abs_{OA} = MAC_{OA} \cdot Mass_{OA}$$

$$= MAC_{BrC} \cdot Mass_{BrC} = MAC_{BrC} \cdot f \cdot Mass_{OA} \tag{2}$$

MAC is the mass absorption coefficient. $f$ is the fraction of OA mass that is BrC. $MAC_{OA}$ is the optical property typically measured in laboratory studies, which includes information on both $MAC_{BrC}$ and the contribution of BrC to OA ($f$). Here, to determine $MAC_{OA}$, we take the OA properties from the Global Aerosol Data Set (GADS) database (Kopke et al., 1997) with updates from Drury et al. (2010), except for the imaginary part of refractive index ($k$).

Little evidence to the contrary, we assume that fossil fuel POA is not absorbing. To date, there are no field observations that indicate POA associated with fossil fuels is light-absorbing (Laskin et al., 2015) except measurements in Beijing (Yan et al., 2017). For biofuel and biomass burning POA, we use the experimental results from Saleh et al. (2014) to parameterize the





imaginary part of the refractive index. In that study, $k$ is related to the BC/OA mass ratio from biofuel and biomass emissions:

$$w = \frac{0.21}{\frac{BC}{OA}+0.07} \qquad (3)$$

$$k_{550} = 0.016 \; \lg(\frac{BC}{OA}) + 0.04 \qquad (4)$$

Where $w$ refers to the wavelength dependence of $k$, $k_{550}$ is the imaginary part of refractive index at wavelength of 550nm. For other wavelengths ($\lambda$), $i$ can be calculated as:

$$k = k_{550}(550/\lambda)^w \qquad (5)$$

The BC/OA emission ratio is associated with both combustion fuel type and burning conditions. In the GFED4s emission inventory, the BC/OA emission ratio ranges from ~0 to 0.23 for biofuel (however the majority of points range lie between
0.06 and 0.16, the $10^{th}$ and $90^{th}$ percentiles) and 0.03 to 0.06 for biomass burning (note that throughout our analysis "biomass burning" refers to open burning and does not include biofuel). These ranges are not large because the variability in burn conditions, which is likely to dominate the variability in BC/OA emission ratio, is not represented. We use global average BC/OA emission ratios in the model for each source: 0.12 for biofuel and 0.05 for biomass burning. This simple assumption reflects the average burning conditions globally but not for specific fires. The assumed BC/OA ratio is further used to derive
the wavelength-dependent $k$. The size distribution of OA is assumed to be log-normal, with a geometric median diameter (GMD) of 180 nm and standard deviation ($\delta$) of 1.6. Based on these values, the MAC of OA at 365 nm is calculated to be 0.70 $m^2g^{-1}$ for biofuel OA and 0.75 $m^2g^{-1}$ for biomass burning OA. The OA-AAE of the 300&600 nm wavelengths pair is 2.6 for biofuel OA and 3.1 for biomass burning OA. These assumptions will be evaluated in the following comparisons with observations. We choose this approach for our model simulation of BrC because relationships between the absorption of OA
and the BC/OA ratio have been confirmed by field measurements (Wang et al., 2016; Gilardoni et al., 2016).

For SOA, we assume that only aromatic SOA absorbs light since experiments show most light-absorbing SOA is related with aromatic carbonyls (Jaoui et al., 2008; Desyaterik et al., 2013; Zhang et al., 2014), and since absorption from biogenic SOA in the field (in the same region and years studied here) has been found to be negligible compared to even mild biomass burning influence (Washenfelder et al., 2015). . We specify the absorption properties of aromatic SOA based on our earlier
study (Wang et al., 2014); these are among the highest values from laboratory experiments (MAC=0.86 $m^2g^{-1}$ at 365nm, Zhang et al., 2013).

All of the above assumptions, for both POA and SOA, are the initial properties for our model simulations. The goal of this study is to investigate whether these assumptions are consistent with the absorption properties in the real atmosphere. To distinguish with other simulations described below, we call this the Base simulation.

**3.3 Chemical aging of biomass burning BrC**

In the Base simulation, we assume that the absorption properties of organic aerosol are fixed. To investigate the influence of chemical aging, we perform additional simulations with assumptions derived from our previous study (Wang et al., 2016). In





that study, we found that the BrC absorption of biomass burning plumes observed at T3 site of the Green Ocean Amazon campaign (GoAmazon2014/5) exhibited a ~1-day photochemical lifetime (in sunlight). This photochemical lifetime is qualitatively consistent with the study of Forrister et al. (2015), who investigate the Rim fires during SEAC4RS. To our knowledge, these two field studies are the only ones to investigate the change in BrC absorption during chemical aging. To

include this aging effect in the model, we assume that the absorption of OA decreases at a rate related to OH:

$$Abs_{BrC,\ t+\Delta t} = Abs_{BrC,\ t} \cdot \exp(-\frac{[OH]\cdot\Delta t}{5\times10^5}) \qquad (6)$$

where $Abs_{BrC,t}$ and $Abs_{BrC,t+\Delta t}$ are the absorption of BrC at time $t$ and $t+\Delta t$ (in days), $[OH]$ is the concentration of OH in molec cm$^{-3}$. As both of these studies found that the absorption did not decrease beyond some minimum threshold, we do not allow absorption to drop below a specified minimum (1/4 of the starting point). We add this scheme to the Base simulation

described above to conduct a model simulation with aging (Base_Age).

## 3.4 Mixing of BC and OA

As discussed in section 1, BC and OA are likely to be internally mixed in a form reasonably well modelled with core-shell morphology (China et al., 2014). This morphology enhances the absorption of BC through lensing, and this enhancement depends upon the absorption properties of the shell material (including BrC). However, this is challenging to represent

accurately, given uncertainties in the coating thickness and composition. Furthermore, considering the low BC/OA emission ratio from biomass burning and biofue, together with the typical coating thickness (Moffet and Prather, 2009; Schwarz et al., 2008; Perring et al., 2016), the majority of OA from these sources is generally externally mixed with BC. Indeed multiple field studies have reported that BC is only present in a few percent of the biomass burning particles, and that the large majority of the emitted particles do not contain BC (Kondo et al., 2011; Perring et al., 2017). Finally, we suggest that as the

observations of mass concentrations are for OA alone (i.e. "externally mixed OA"), these should be compared with externally mixed OA in the model. Therefore, we treat BC and OA as externally mixed in our simulation. We apply a constant absorption enhancement for BC (1.1 for fossil BC, 1.5 for biofuel/biomass burning BC), as described in Wang et al. (2014), based on a series of laboratory and field observation, regardless of whether the coating shell absorbs light or not. As a result, this value likely represents some average state which includes the influence of BC-OA internal mixing. Our

assumption of externally mixed OA with an associated absorption enhancement for BC may overestimate OA absorption since the OA which coats BC is double-counted; however given that the majority of the OA is likely externally mixed this overestimate in absorption is modest, and likely negligible for air masses influenced predominantly by biofuel and biomass burning sources.

## 4 Comparing simulated BrC to aircraft observations

In this section, we evaluate our assumptions for BrC by comparing the GEOS-Chem nested model simulations with aircraft observations from the DC3 and SEAC4RS campaigns. The region included in the analysis is the central and southeast US,





which is shown in Figure 1. We focus on this subset of the measurements because: 1) aircraft measurements from both DC3 and SEAC4RS cover this region; and 2) the emissions inventories for this region have been evaluated by a series of SEAC4RS studies (resulting modifications described in Section 3.1). Before evaluating the simulation of OA absorption, we first need to explore the fidelity of the simulation of OA mass.

## 4.1 DC3 Campaign

We first compare the Base simulation to observations. Figure 2 compares the median vertical profile of modelled sulphate, BC, and OA mass concentrations with the DC-8 aircraft measurements during the DC3 campaign. Our simulation reproduces the median vertical distribution of observed sulphate and BC, but underestimates OA by about a factor of two at low altitudes (< 3 km). To investigate the source of this bias, we show all the observed 1-minute-average data points together with model results as a "points-to-points" plot in Figure 3. The model reproduces the BC observations (normalized mean bias (NMB) = -5%) except for some occasional peaks, which are challenging to capture given the limitations of model temporal and spatial resolution. We note here that the model skill in capturing this variability improves in the nested grid (R=0.54) compared to the global $2° \times 2.5°$ grid (R=0.48), with little change in NMB. In comparison, the bias in the simulation of OA is much larger, with an overall NMB of -45%. The unbiased simulation of sulphate and BC suggests that the model generally captures the transport, deposition, and primary emissions (fossil, biofuel and biomass burning) of aerosols. Therefore, the underestimate of OA is more likely associated with biased emission factors for POA and/or an underestimate of SOA. A key question is whether this bias is associated with absorbing or non-absorbing sources of OA.

According to the emission inventories used here, biofuel contributes very little OA in this region (< 3% of the total POA source during the campaign). This is consistent with a negligible demand for heating during spring and summer in the Southeast US. We therefore conclude that it is highly unlikely that the substantial OA underestimation identified in Figures 2c and 3b is associated with biofuel sources. To investigate whether an underestimate in fire emissions contributes to the bias, we also show the measured acetonitrile ($CH_3CN$) concentrations in Figure 3c. Acetonitrile is a tracer for biomass burning and biofuel emissions (Andreae and Merlet, 2001). We calculate the hourly correlations between $CH_3CN$ and OA to help to identify whether the OA during plumes are associated with fires. When a $CH_3CN$ peak and high $CH_3CN$-OA correlation are both observed, we can be confident that biomass burning dominates the sources of OA. We observe two such periods (BP1 and BP2), which are shown with green shading in Figure 3. BP1 is a period with a series of $CH_3CN$ peaks measured in the central US. The correlations between $CH_3CN$ and OA are continuously high ($R^2 = 0.5 - 0.9$) throughout the period. Both modelled BC and OA are dominated by biomass burning and enhanced during BP1, but underestimate the measurements. This suggests that the model does not capture the strength of these plumes. The simulated mass of BC from fires needs to increase by 130% to match observations. This bias could be associated with transport (including excessive dilution), inaccuracies in the amount or intensity of burning in the emissions inventory or in the emission factor for fire sources of BC. In contrast, the mass of biomass burning OA needs to increase by 210% to match the observations; this is ~80% more than for BC. BP2 is a 1-hour period dominated by a biomass burning plume observed in the southeast US, with





very high $CH_3CN$-OA correlation ($R^2 = 0.84$). The model is able to represent the BC concentrations during this period quite well (<10% underestimate), though if we attribute this entire bias to the biomass burning source, it implies a 36% increase in that source. Similarly, the mass of biomass burning OA needs to increase by 145%, which is also ~80% more than for BC (similar to BP1). Since the influence from transport and errors in fuel burned should be very similar for BC and OA, the

higher bias in simulated OA suggests that either the biomass burning BC/OA emission factor is overestimated, or that biomass burning constitutes a large source of SOA which has been neglected in the model. If all of the 80% difference is due to the overestimate of BC/OA emission ratio, the BC/OA emission ratio would need to be reduced to 0.027 to meet the observations; this value is lower than the emission factor for any fire type in the GFED4s emission inventory. It is therefore unlikely that this difference can be attributed entirely to an overestimate of the BC/OA emission factor. A number of studies

have explored the formation of SOA in biomass burning plumes. Yee et al. (2013) conduct photo-oxidation experiments in their chamber, and find that the formation of SOA from oxidation of phenol, guaiacol and syringe can be larger than 25% of the co-emitted biomass burning POA. Ortega et al. (2013) investigate the biomass burning smoke from fuels combusted during the FLAME-3 study and find that the net increase in mass due to biomass burning SOA is $42 \pm 36\%$ of the biomass burning POA. However, compared to the laboratory studies, aircraft field measurement show much less SOA formation.

Cubison et al. (2011), Jolleys et al. (2012), and Shrivastava et al. (2017) have reviewed all aircraft field studies of SOA formation in BB plumes, and found that the increase in total OA from SOA production is most often undetectable, with a smaller fraction of the cases showing increases or decreases with aging, which are a small fraction of the initial POA. Some studies have included simple biomass burning SOA schemes in their models. Hodzic and Jimenez (2011) assume a simplified biomass burning SOA scheme in the CHIMERE model (VOC is oxidized by OH with a constant rate); their

simulation estimates that biomass burning SOA contributes 11% of the total SOA in Mexico City. Kim et al. (2015) used the same scheme for the SEAC4RS period and concluded that biomass burning SOA contributed 1% of the OA in this region, and comparable to 10% of the biomass burning POA. Therefore it is unlikely that the majority of the 80% bias can be attributed to SOA formation from fires. The small bias in the simulation of BC during BP2 suggests that the emission factor for BC from biomass burning is not substantially biased. Rather, it is likely that the bias in BC (particularly in BP1) results

from an underrepresentation of total emissions from these fires. In both BP1 and BP2, after adjusting both BC and OA mass concentrations upwards to eliminate this bias, we still need to increase OA by an additional 80% to account for the underestimate of either the POA emission factor or biomass burning SOA. This 80% represents the upper limit on missing OA associated with biomass burning, given that other sources likely contribute to background concentrations.

Based on above analysis, we first increase the biomass burning OA mass by 210% and 145% during BP1 and BP2,

respectively, to fix the model bias associated with these specific fire plumes. We then increase the biomass burning OA mass by 80% for all the remaining data (including background biomass burning OA), likely to account for the bias from the POA emission factor and any missing biomass burning SOA. This modified simulation of OA mass (referred to as FixBB) reflects the highest possible biomass burning contribution. In Figure 2c, this modified model (same as Modified_Age, described at the end of this section) still underestimates the vertical profile of OA mass. This underestimate can be observed as





underestimated OA peaks in Figure 3b but not in the corresponding BC concentrations in Figure 3a, and is therefore unlikely to be related to combustion sources (fossil fuel, biofuel, or biomass burning). This suggests that the remaining underestimate of OA is related to anthropogenic and/or biogenic secondary sources, which are not a source of BC. This is consistent with previous work which suggests a general underestimate of SOA in the GEOS-Chem simulation (Heald et al., 2011).

Furthermore, OA absorption is not enhanced during these peaks, suggesting that this SOA is not strongly absorbing and these biases are not relevant to our analysis of absorption which follows.

Figure 2d compares the simulated median vertical profile of OA absorption with measurements from the DC3 campaign. Note that this vertical profile may not be entirely representative since there are very few data points (< 10) available at some altitudes. It is also important to note that the OA particle size may differ between model and observations. The size

distribution assumed in the model is for fine mode particles, and would not include the absorption from coarse mode OA (>1μm diameter). The biomass burning source contributes ~90% of the total absorption from OA in our simulation. The model captures the OA absorption variation at low altitudes even with the underestimate of OA mass shown in Figure 2c, but underestimates the absorption at high altitudes. At altitudes above 10 km, the observations show abnormally high OA absorption considering the correspondingly low sub-micron OA mass. The model fails to capture these high values even by

applying the highest absorption properties from laboratory studies for OA. Zhang et al. (2017) analyzed the inflow and outflow of OA absorption during DC3 and conclude that the high absorption aloft may relate to coarse mode OA or OA formation during convective transport. A number of studies also suggest that aqueous-phase chemistry in cloud droplets at high altitudes can produce absorbing OA (Ervens et al., 2011; Desyaterik et al., 2013). These sources of OA are not included in our simulation and given the limited observational constraints provided by this dataset, we do not consider this data

further in our study, but agree with Zhang et al. (2017) that further investigation of high altitude BrC is needed. The absorption enhancement in both observations and model around 4.5km is related to biomass burning; this is confirmed by elevated observed acetonitrile concentrations at this altitude. The high concentrations at this altitude are influenced by the fire plumes during BP1.

To investigate the absorption properties of biomass burning OA, we select the data during BP1 as we are confident that

nearly all of the OA absorption is related to biomass burning in this period. Figure 4 compares OA absorption from simulated biomass burning OA with measurements during BP1. The Base modeled biomass burning OA absorption is moderately correlated (R = 0.56) with the observations but overestimates them by ~50%. This overestimation increases to more than 3 times after the underrepresentation of fire OA is corrected (FixBB). This suggests that the model assumption for $MAC_{OA}$ is too high. Given this overestimate, we perform an additional simulation which includes photochemical

"whitening" of BrC (described in Section 3.3, FixBB_Age); the results are shown as green points in Figure 4. By applying the aging scheme, the correlation between modeled and observed absorption increases (R = 0.62), and the model is brought into much better agreement with observations (NMB=+30%). We note that if the 80% increase in biomass burning sourced OA included in FixBB is attributed solely to an overestimate of the BC/OA ratio (which we previously note seems unrealistic) this would imply an 8% decrease in the MAC following the Saleh et al. (2014) parameterization used here. This





suggests that the overestimate of our initial assumption of MAC$_{OA}$ is more likely due to the difference between laboratory experiments and the mean condition (e.g. fuel types, burning temperature, etc.) for the United States, than the overestimate of global BC/OA ratio used in our simulation.

Figure 4 suggests that our initial assumption of MAC$_{OA}$ for biomass burning is too high; it must be reduced by 24% when including the whitening process (76% if not considering aging) to match the observations in BP1. At 365 nm, the best MAC$_{OA}$ to represent the measurements is 0.57m$^2$/g with aging (this is the MAC for freshly emitted OA) and 0.18m$^2$/g without aging. The 0.18 m$^2$/g value is around the lower end of previous experimental studies, whereas 0.57m$^2$/g falls close to the median of previous experimental studies. This supports the idea that an aging process may be important for simulating OA absorption. Given that there are only three OA absorption measurements available during BP2, we cannot repeat this analysis for BP2. For the other periods not dominated by biomass burning, the correlation between modeled and observed OA absorption is very low (R < 0.1). OA absorption is typically lower during these periods and represents a mix of biomass burning and biofuel influences, the combination of which reproduces the magnitude of observed OA absorption.

After applying a series of new model assumptions, which include increasing fire OA mass, decreasing the biomass burning MAC$_{OA}$, and adding an aging scheme, we conduct a new simulation (Modified_Age). The simulated vertical profile of OA absorption in this simulation is nearly identical to the BASE model, however, this new simulation also captures the OA mass from biomass burning plumes. Since the observational constraints on MAC$_{OA}$ and the aging scheme only affect absorption but not aerosol mass, the simulated OA concentrations in FixBB, FixBB_Age and Modified_Age are the same.

## 4.2 SEAC4RS Campaign

The SEAC4RS campaign offers us the opportunity to test our updated simulation developed based on DC3 measurements with a new dataset. Figures 5 and 6 show the vertical profiles and points-to-points plot for DC-8 aircraft measurements during SEAC4RS. Similar to DC3, our model generally captures the median vertical profile of sulphate (Figure 5a) and BC (Figure 5b). During SEAC4RS, biogenic SOA constitutes a much larger source of OA in the model (compared to DC3). Consistent with DC3, the Base simulated OA absorption captures the observations at low altitudes but is too low at high altitudes. The observed absorption at high altitudes is much lower than observed in DC3. Zhang et al. (2017) suggest that this is because measurements during SEAC4RS are less influenced by convection than DC3, thus there may be less secondary formation of BrC during convective transport. Note that there are very few data points available at altitudes above 4 km. The Base model underestimates the OA mass observations, but with a much lower bias (~50%) than seen during DC3. Similarly, the model bias for BC is modest (NMB=-30%). There is therefore weaker evidence for missing or underestimated fire activity in the GFED4s inventory during SEAC4RS. Furthermore, Figure 6c shows that there are no coincident peaks with both elevated CH$_3$CN and CO correlated each other. In our selected region during SEAC4RS there is no clear period which is dominated by biomass burning. The Rim fires occurred on 26-27 August 2013. During this period, CO is underestimated by more than 400% in the model, which indicates that the model fails to reproduce the fire plumes from the





Rim fires. However, as shown in Figure 1, these measurements are located around the northwest US and are not included in our analysis.

When applying the same modified model assumptions constrained from DC3 (Modified_Age simulation), the model simulation of both observed OA concentrations (Figure 5c) and absorption (Figure 5d) during SEAC4RS improves. The mean bias between model and observation decreases from -42% to -28% for OA mass concentrations, and from +32% to -23% for OA absorption in the Modified_Age simulation (considering only altitudes below 4km where there are sufficient measurement data points). This confirms that the modifications applied based on the DC3 campaign in 2012 are generally appropriate for this region.

Washenfelder et al. (2015) analyzed measurements of OA absorption at a surface site within the study region (central Alabama) and during a similar time period (June 2013) of SEAC4RS. They found that most of the OA absorption was associated with biomass burning with little contribution from SOA, consistent with our analysis of the aircraft data. They suggest a biomass burning $MAC_{OA}$ of $1.35 m^2 g^{-1}$, much higher than ours. However, as their site was rarely affected by biomass burning (~6% of all OA), the identification of biomass burning OA absorption properties from this site is challenging and may not be regionally representative.

### 4.3 Recommendations for OA absorption properties

Although the assumption of a relationship between BrC absorption and the BC/OA ratio, which is applied in our simulations, has been observed in several studies, including both laboratory (Saleh et al., 2014; Pokhrel et al., 2017) and field measurements (Wang et al., 2016; Gilardoni et al., 2016), the specific relationship (e.g. slope) differs among these studies. Based on the above analysis, our assumed MAC for fresh biomass burning OA at 365nm (based on Saleh et al., 2014) needs to be decreased by 24% to reproduce the observations from DC3 and SEAC4RS (when including an aging scheme). As discussed in Section 4.1, this may represent the difference between the laboratory experiments and the mean condition for the United States rather than that U.S. fire emissions exhibit a lower BC/OA ratio than the mean values used in our simulation. As a result, we retain the absorption wavelength dependence based on Saleh et al. (2014), but decrease the $MAC_{OA}$ by 24% in the model. Our recommended $MAC_{OA}$ for biomass burning is therefore $0.57 m^2 g^{-1}$ at 365nm, $0.33 m^2 g^{-1}$ at 440nm and $0.15 m^2 g^{-1}$ at 550nm with the suggested aging scheme described in Section 3.3. We assume that biomass burning SOA is equally absorbing as primary OA from biomass burning. All of these numbers can be translated to the form of $MAC_{BrC}$ if the contribution of BrC to OA is known or specified. For example, the $MAC_{OA}$ of $0.57 m^2 g^{-1}$ is equivalent to a $MAC_{BrC}$ of $1 m^2 g^{-1}$ with BrC contribution of 57% to total OA. 2012 and 2013 were not exceptionally low or high fire years in the United States. During DC3 and SEAC4RS, fires in the United States (e.g. 12586 fires in June 2012, data from www.globalfiredata.org) were somewhat more frequent than the last 10-years average (e.g. average 9831 fires in June). The difference between 2012 and 2013 and 10-years average emissions in our research region during the measurement period are not large (22% and 34% higher in 2012 and 2013 respectively, compared to the 10-year average). This suggests that our



conclusions based on the constraints from these two campaigns can be generalized to other biomass burning seasons in United States.

The spring and summer in the southeast U.S. are not substantially impacted by biofuel emissions, therefore the measurements during the DC3 and SEAC4RS campaigns are not suitable for evaluating the absorption from biofuel OA.

Given that we see no model bias when biofuel influence exceeds the biomass burning influence (typically during low absorption background OA periods), we retain our assumptions in Section 3.2 for biofuel OA. Therefore, our recommended value for biofuel $MAC_{OA}$ is $0.7m^2g^{-1}$ at 365nm, $0.45m^2g^{-1}$ at 440nm and $0.23m^2g^{-1}$ at 550nm. We assume that there is no "whitening" of biofuel OA with aging given that, to date, there is no field evidence to support this. These assumptions require further testing against measurements with significant biofuel influence.

During both DC3 and SEAC4RS, anthropogenic SOA contributes very little absorption in the model (~4% in DC3 and < 1% in SEAC4RS4) despite the fact that we apply upper-limit assumptions regarding the absorption properties of SOA. In our analysis of DC3, there remain several underestimated OA mass peaks even after increasing biomass burning OA mass. These peaks are likely due to secondary sources. During SEAC4RS, there are also some peaks with substantial simulated biogenic SOA; observed absorption is not elevated during these peaks. Therefore, we conclude that the absorption from

biogenic and anthropogenic SOA is negligible in the Southeast U.S, consistent with SOAS results (Washenfelder et al., 2015). This may not be true in other regions.

Using the above model configuration (Modified_Age simulation), the model is able to reproduce the vertical profile of OA absorption during DC3 and SEAC4RS except for altitudes above 10 km. Our optimized $MAC_{OA}$ is lower than all of the previous BrC model studies. Feng et al. (2013) assume that 66% of the OA from biofuel and biomass burning is BrC. They

applied 2 different sets of assumptions for the absorption properties of BrC: a moderately absorbing BrC with MAC = $0.63m^2g^{-1}$ at 450nm, and a strongly absorbing BrC with MAC = $1.6m^2g^{-1}$ at 450nm. These numbers are 0.41m²/g and 1.06m²/g when transferring $MAC_{BrC}$ to $MAC_{OA}$, which are similar to our biofuel assumptions but much higher than our biomass burning assumptions. Jo et al. (2015) assume different BrC to OA contributions for different biomass burning and biofuel fuels, resulting a range of $MAC_{OA}$ to be 0.65–5.01 $m^2g^{-1}$ at 365nm, higher than our assumptions. The assumptions of

Saleh et al. (2015) and Wang Q. et al. (2016) are also based on the laboratory results from Saleh et al. (2014) but use the modeled BC/OA mass concentration ratio instead of emitted BC/OA ratio. The modeled BC/OA mass concentration ratio reflects the mixed contribution of various sources and the effect of differential removal during transport; it is not equivalent to the initial emission properties of carbonaceous aerosol. They assume all OA from biofuel and biomass burning is BrC and apply a MAC = 2.5m²/g for biofuel OA and 3.1m²/g for biomass burning OA at 550nm. These values are more than 4 times

higher than ours and higher than any of the previous experimental studies. Although some of these modelling studies compare their simulated total AAOD with observations, none evaluates their assumptions with direct BrC absorption measurements.

Simulating the whitening process of BrC as we do in our Modified_Age simulation comes at a computational cost of adding extra species or tracking absorption in a model, something which may not be practical for all 3D models. If we neglect the



whitening of BrC with aging in our simulation, we must further reduce the $MAC_{OA}$ for biomass burning to match the observational constraints. In this case, we estimate an $MAC_{OA}$ for biomass burning of $0.18 m^2 g^{-1}$ at 365nm, $0.11 m^2 g^{-1}$ at 440nm and $0.05 m^2 g^{-1}$ at 550nm; we call this the Modified_Simple simulation. Another approach to simplify the whitening process for models may be to apply constant whitening factors with altitude to the simulated absorption; the consistency of these factors may require additional observational support.

## 5 Global Implications

The model assumptions applied in Modified_Age and Modified_Simple are constrained from US fires observed during DC3 and SEAC4RS. We assume that such constraints are generalizable though the combustion conditions may differ in other regions. We test this assumption in Section 5.1 by comparing our global simulation with AAOD observations outside of the United States. In this analysis, we conduct Modified_Age and Modified_Simple simulations with a horizontal resolution of $2°\times2.5°$, using 2014 meteorology, and 10 year averaged biomass burning emissions (2005-2014).

## 5.1 Surface absorption and AAOD of BrC

We use the results from Modified_Age to conduct the analysis in this section. Both of the Modified_Age and Modified_Simple simulations are optimized to meet the observational constraints, and therefore exhibit very similar surface absorption.

Figure 7 shows the global distribution of surface OA absorption contributions (= BrC absorption contribution) to total absorption from aerosols at 370nm. The modeled contribution ranges from 3% to 58% globally, with an average value of 21%. The circles in Figure 7 show the BrC absorption contributions from observations at 8 surface sites. These data are derived from multiple-years' multiple-wavelength absorption measurements from aethalometers (AE, Magee Scientific, http://www.mageesci.com), using a BC-BrC absorption separation method. Details of this methodology and the specific datasets can be found in Wang et al. (2016). Although the model assumptions were optimized based on measurements in the U.S., the model is able to represent the BrC absorption contribution at many sites in other regions. The simulated contributions reflect 10-years averaged values over all seasons, but the measurements over these 8 sites are not continuous and usually cover several months in a year. Therefore, this is not an exact comparison between the model and these measurements. However, the BrC contributions in both measurements and model in these locations range over 15-40%, which suggests that the modeled surface BrC absorption contribution does not have significant bias in most regions.

Figure 8 shows the global distribution of simulated column BrC-AAOD and the contribution of BrC-AAOD to total AAOD at 440nm.The BrC-AAOD at 440nm ranges from $\sim10^{-5}$ to 0.045, with a global mean of 0.001. BC still dominates the total AAOD in most regions. The contribution of BrC-AAOD to total AAOD at 440nm ranges from 10% to 60%, with the global mean of 46%.The Aerosol Robotic Network (AERONET) provides a worldwide measurement network of AAOD at four wavelengths (440, 675, 870, and 1020 nm). However, several shortcomings limit its use for constraining modeled BrC-



AOOD, which include the uncertainties on AERONET retrievals, possible inconsistencies between assumptions in the retrieval scheme and our model, poor data availability, no data at low wavelengths where BrC dominates absorption, and the influence from dust. Details of the processing and the uncertainty issues surrounding AERONET AAOD are discussed in Wang et al. (2016). When assuming only BC absorbs light, the modeled AAOD has a moderate correlation with AERONET

AAOD at 440nm (R = 0.53). This correlation is smaller than that at 675nm (R=0.59) where OA contributes nearly no absorption. After including the absorption from BrC, the correlation of AAOD at 440nm increases to R = 0.60; this increase in model skill qualitatively supports our description of BrC absorption.

These comparisons suggest that our simulation, optimized based on observations in the United States, is not obviously biased in other regions of the world. However, we emphasize that we have developed a simple approach to modeling BrC and more

observations are needed to refine this simulation for other regions where sources and optical properties may differ.

## 5.2 Estimating the direct radiative effect (DRE) of BrC

Figures 9a and c show the DRE of total OA from the Base and Modified_Age simulations. The global mean value of all-sky DRE is -0.290 Wm$^{-2}$ and -0.350 Wm$^{-2}$ in Base and Modified_Age, at the top of atmosphere. This number is -0.392 Wm$^{-2}$ when assuming OA does not absorb light. Therefore the global mean absorption DRE from OA (BrC) is estimated to be

+0.102 Wm$^{-2}$ and +0.042 Wm$^{-2}$ in Base and Modified_Age. In the Modified_Age simulation, biofuel and biomass burning sources contribute 60% and 40% respectively to the global absorption DRE of BrC. The absorption DRE of BrC from our best (Modified_Age) simulation (+0.042 Wm$^{-2}$) is about 25% of the DRE from BC (+0.17 Wm$^{-2}$). The aging process significantly impacts our estimate of the absorption DRE; the global absorption DRE is 43% higher when using the same optical assumptions but excluding the aging scheme in Modified_Age. We also find that the global mean absorption DRE is

very similar using the Modified_Simple scheme (+0.045 Wm$^{-2}$) or the Modified_Age scheme, both of which are observationally constrained.

The DRE could be underestimated due to two reasons: First, we attribute all the mass bias of OA to biomass burning OA during fire plumes without considering other sources. This may overestimate the contribution of biomass burning in our analysis, thereby underestimating the MAC of biomass burning OA when constrained by absorption observations in fire

plumes. Using a potentially underestimated MAC globally could result in the underestimate of the global DRE. Second, we neglect the OA absorption at high altitudes (> 10km). Zhang et al. (2017) suggest that this contributes a local DRE of 0.65±0.34 Wm$^2$. However it is unclear how important this convectively-formed BrC is globally, therefore we neglect it here, implying that our estimate of absorption DRE of BrC is a lower estimate.

In contrast, the DRE could also be overestimated for two reasons: First, If we assume that biofuel BrC is subject to the same

aging process as biomass burning BrC, the global absorption would be DRE 40% lower than our estimate. Second, we assume that BrC is completely externally mixed with other aerosols. This will overestimate the absorption since the BrC coated on BC is also counted (via an absorption enhancement factor). Assuming the shell thickness is ~60% of the core radius (observed in field measurements, Cross et al., 2010; Shirawa et al., 2010), and all coated material is BrC for



biofuel/biomass burning related BC, the absorption DRE of BrC will be ~15% lower. This effect is likely even smaller given that BrC may contribute little to the coating material compared to non-absorbing OA and nonorganic aerosols.

Our estimate of BrC DRE is lower than previous studies which have not been evaluated against direct measurements of BrC absorption. Saleh et al. (2015) apply BrC absorption properties based on the modelled BC/OA ratio and estimate the absorption DRE from BrC to be +0.12 Wm$^{-2}$ to +0.22 Wm$^{-2}$. Jo et al., 2016 use the modified combustion efficiency (MCE, a function of CO/CO$_2$) to determine BrC absorption, and estimate the absorption BrC DRE to be +0.11 Wm$^{-2}$. These values are similar to the absorption DRE of BrC estimated from our Base simulation, prior to optimization against observations. In this study, we do not estimate the DRF, which is the difference between pre-industrial and present-day DRE, given the challenges in identifying the anthropogenic fraction of biomass burning emissions. Several previous studies report the absorption DRF for BrC: +0.04 Wm$^{-2}$ to +0.11 Wm$^{-2}$ by Feng et al. (2013) and +0.22 Wm$^{-2}$ to +0.57 Wm$^{-2}$ by Lin et al. (2014).

**6 Conclusions**

We use the GEOS-Chem model coupled with RRTMG model to investigate the mass optical properties and direct radiative effect of brown carbon (BrC). Our model assumptions for the optical properties of BrC are based on the laboratory study of Saleh et al. (2014) and constrained by the aircraft measurements from the DC3 and SEAC4RS campaigns in the US.

Our model captures the magnitude and vertical distribution of sulphate and BC mass concentrations during both DC3 and SEAC4RS. However, the model underestimates the OA mass concentrations in both campaigns. By analyzing the fire plumes in the observations, we find the biomass burning OA is likely to be underestimated by 80% due to the bias in OA emission factors and missing biomass burning related SOA. After fixing the OA mass from biomass burning, our model is able to represent the variation of OA absorption in fire plumes, but substantially overestimates the magnitude. Applying an aging scheme where OA photochemically whitens further increases the correlation between modeled and observed absorption and decreases the model bias. These comparisons, suggest fire emissions are characterized by an MAC$_{OA}$ of 0.57m$^2$/g at 365nm, 0.33m$^2$/g at 440nm and 0.15m$^2$/g at 550nm, which decreases with aging. The optical properties for biofuel emissions are not well constrained by these datasets, and we retain our original assumptions based on Saleh et al. (2014) with biofuel MAC$_{OA}$ of 0.7m$^2$/g at 365nm, 0.45m$^2$/g at 440nm and 0.23m$^2$/g at 550nm. Using these assumptions, we estimate a global mean top-of-the-atmosphere DRE of -0.359 Wm$^{-2}$ for OA, and an absorption DRE of +0.042 Wm$^{-2}$ for BrC, all-sky conditions. These properties and the resulting estimated DRE are lower than values in previous modeling studies, however, none of these studies have been constrained by or evaluated against direct BrC absorption measurements. The absorption DRE of BrC may be larger than estimated here if the in-cloud heterogeneous source of upper altitude BrC observed by Zhang et al. (2017) is globally significant.

Although the model can reproduce the aircraft observations from DC3 and SEAC4RS when using the above model configuration, further studies, especially global, direct measurements, are necessary to build a credible simulation of BrC.





First, current emission inventories do not provide enough information to accurately apply combustion condition based BrC absorption properties. Emission measurements representative of varying burning conditions as well as different fuel types are needed. Second, we extend the model assumptions constrained from regional observations (mainland US) to a global simulation. It is not clear whether BrC properties are consistent world-wide. Third, more studies are required to investigate

the contribution of biomass burning SOA and its absorptivity. Fourth, the absorption assumptions for biofuel OA must be more thoroughly evaluated. It is also not clear whether the whitening process also affects the absorption of biofuel OA. Last, our simulations do not include OA absorption from fossil fuels. Fossil OA has only been identified as light-absorbing in Beijing (Yan et al., 2017). However, applying this assumption for fossil OA world-wide in the model would substantially increase background OA absorption, leading to a considerable model overestimate of OA absorption observed during aircraft

campaigns (Section 4) and OA absorption contributions from surface sites (Section 5.1). Therefore, to further constrain the global impacts of BrC, additional field measurements representative of various source influences (fossil, biofuel OA, SOA) are required.

### Acknowledgements

This work was supported by EPA (RD-83503301-0) and NOAA (NA16OAR4310112). RJW was supported through a NASA Radiation Sciences Program grant NNX14AP74G. PCJ and JLJ were supported by NASA NNX15AT96G. JPS and AEP were supported by the NOAA Atmospheric Composition and Climate Program, the NASA Radiation Sciences Program, and the NASA Upper Atmosphere Research Program.

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



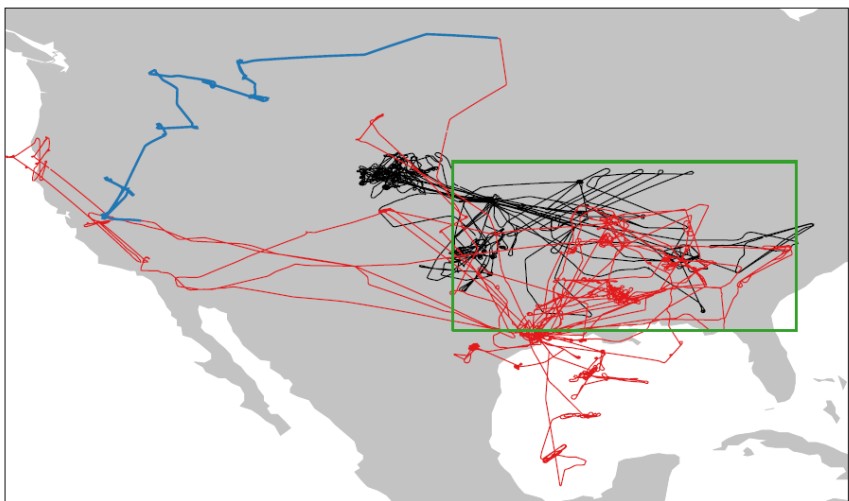

Figure 1. The flight tracks during the DC3 (black) and SEAC4RS (red) campaigns in 2012 and 2013, respectively. The blue tracks indicate the SEAC4RS data influenced by Rim fires on August 26-27. The green box indicates the region of focus for our analysis (see Section 4 for details).





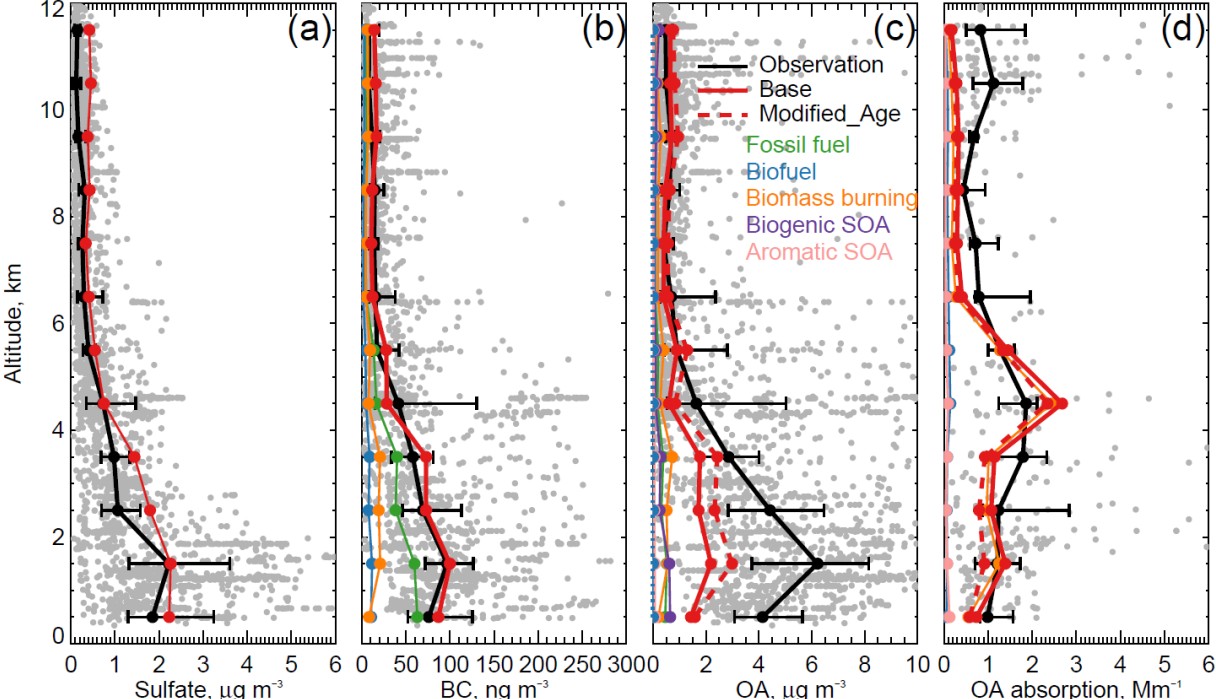

Figure 2. The median vertical profile of (a) sulfate, (b) BC, (c) OA mass concentration and (d) OA absorption, shown in 1
km bins, from the DC-8 aircraft measurement during DC3 campaign in the region shown in Figure 1. Observations (black)
are compared to the Base simulation (red) and source-specific contributions to that simulation, as well as to the optimized
Modified_Age simulation (red dashed). Error bars show the 25th and 75th percentiles of measurements in each vertical bin.
Gray points show the original measurements (1-minute averaged values for a, b and c, 5-minutes averaged values for d). The
ranges of x-axes are set to emphase the vertical profile, so several data points higher than the maximum values of x-axes are
not shown. Details regarding the model simulations of Base and Modified_Age can be found in Sections 3.2 and 4.1.



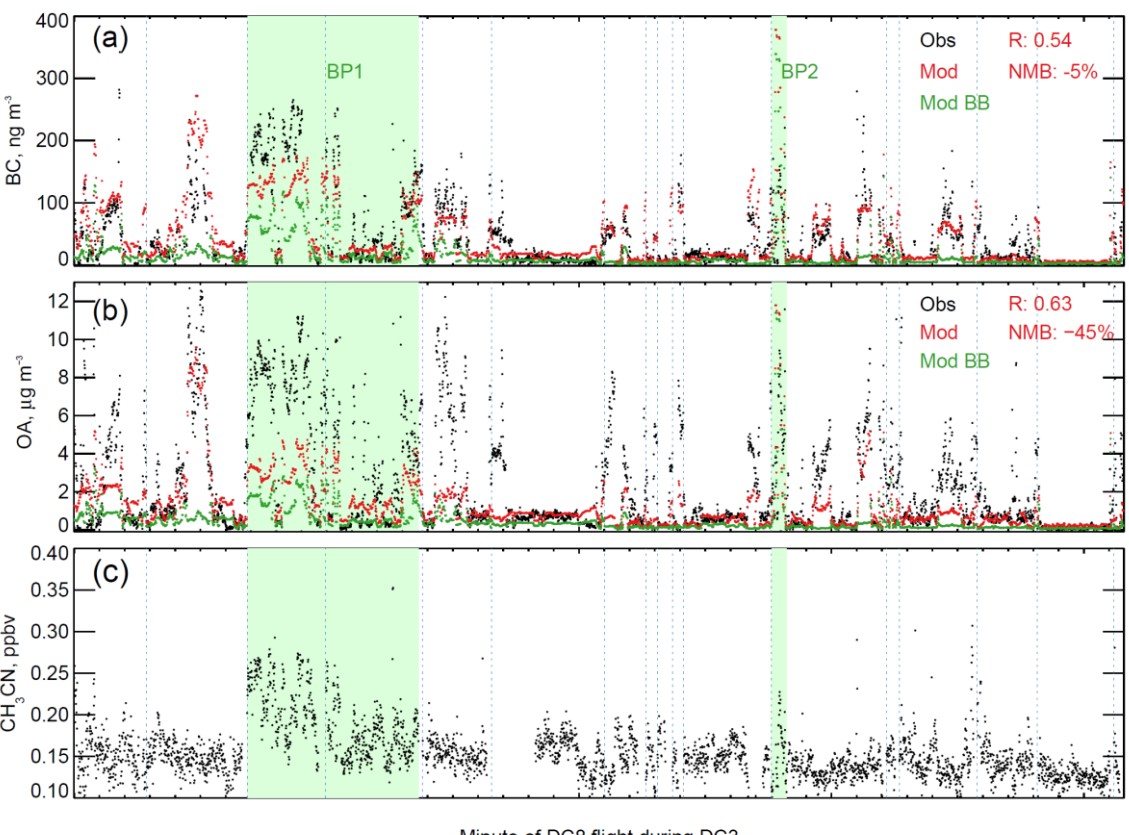

Figure 3. Points-to-points comparison between 1-minute averaged observed (black) and simulated (red) (a) BC and (b) OA made aboard the DC-8 aircraft during the DC3 campaign in the region shown in Figure 1. The simulated total mass concentrations (red) as well as mass concentrations associated with biomass burning only (green) are from the Base simulation. The observed concentrations of acetonitrile are also shown (c). The blue dashed lines separate different flights. The green shading indicates two biomass burning dominated periods (BP1 and BP2, discussed in Section 4.1).





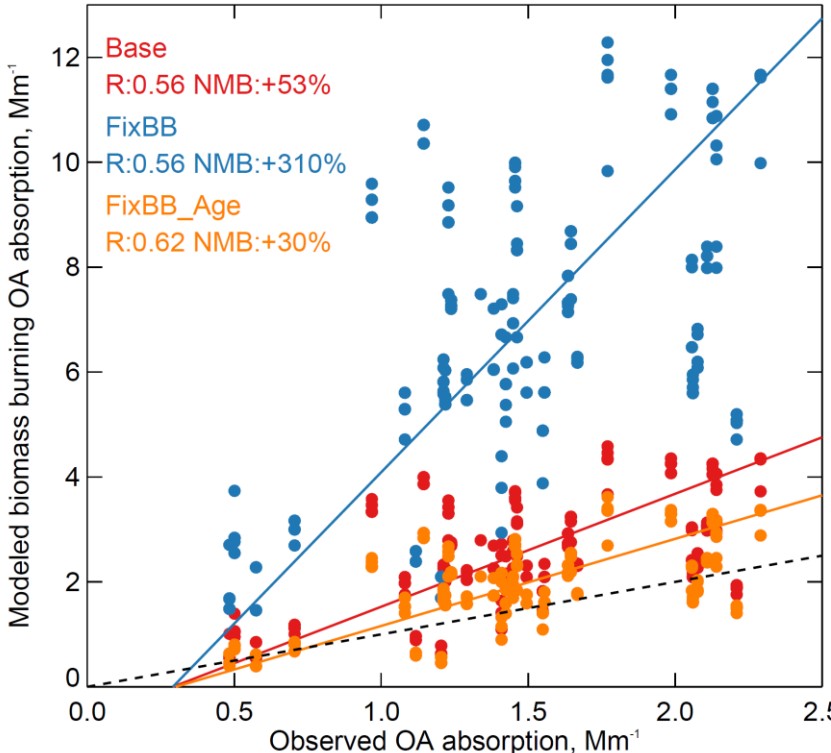

Figure 4. Correlation between observed and modelled OA absorption during the BP1 interval (see Figure 3) of the DC3 campaign. The 1-to-1 line is shown as a dotted black line, the best-fit lines are shown as solid lines. NMB: normalized mean bias between the simulation and observations. Details of the model simulations can be found in Section 4.1.





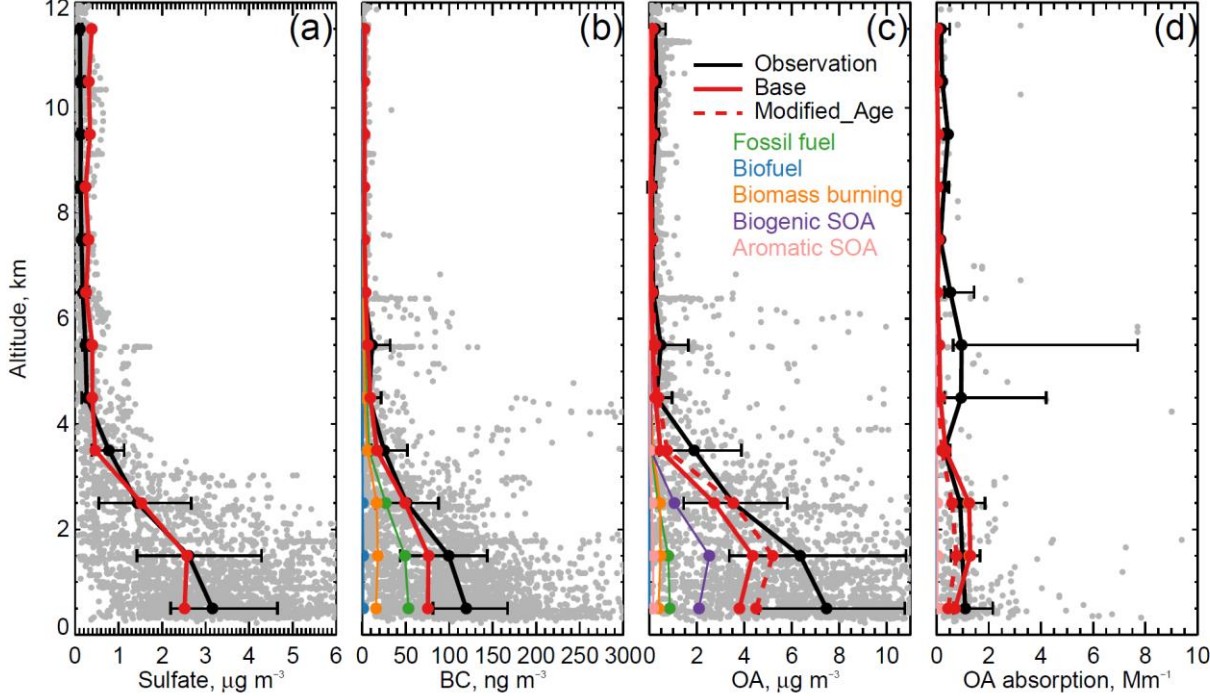

Figure 5. The median vertical profile of (a) sulfate, (b) BC, (c) OA mass concentration and OA absorption (d), shown in 1 km bins, from the DC-8 aircraft measurement during SEAC4RS campaign in the region shown in Figure 1. Observations (black) are compared to the Base simulation (red) and source-specific contributions to that simulation, as well as to the optimized Modified_Age simulation (red dashed). Error bars show the 25[th] and 75[th] percentiles of measurements in each vertical bin. Gray points show the original measurement data points (1-minute averaged values for a, b and c, 5-minutes averaged values for d). The ranges of x-axises are set to emphasize the vertical profile, so several data points higher than the maximum values of x-axes are not shown. Details of model simulations of Base and Modified_Age can be found in Section 3.2 and 4.1.





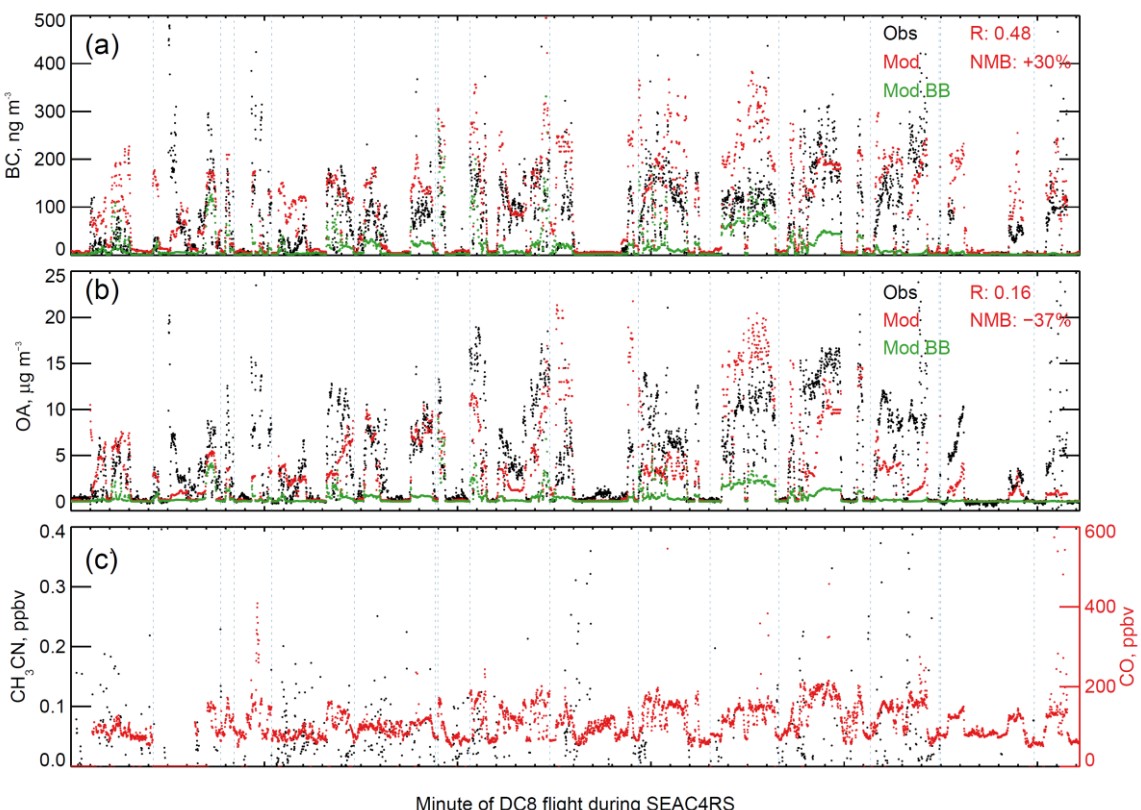

Figure 6. Points-to-points comparison between observed and modelled (a) BC and (b) OA made aboard the DC-8 aircraft during the SEAC4RS campaign in the region shown in Figure 1. The modelled total mass concentrations (red) as well as mass concentrations associated with biomass burning only (green) are from the Base simulation. The observed concentrations of acetonitrile and CO are also shown (c). The blue dashed lines separate different flights.



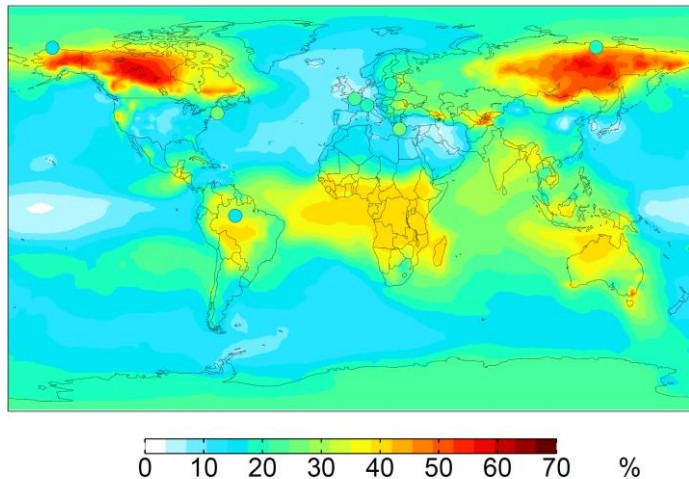

Figure 7. Global distribution of simulated BrC absorption contribution to total aerosol absorption at the surface for 2014. Results are from the Modified_Age simulation. The circles show the retrieved results from multiple-wavelength absorption measurements at 8 surface sites (see Section 5.1 for details).



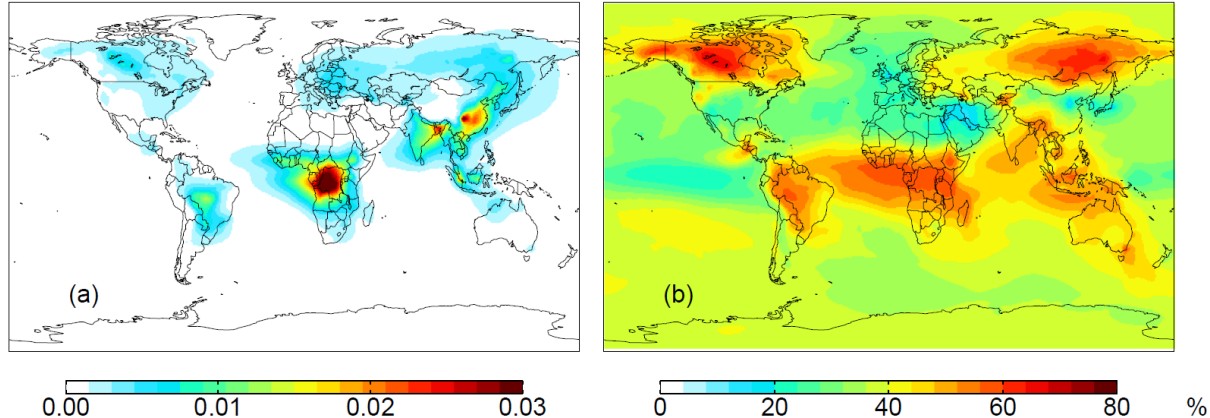

Figure 8. Global distribution of simulated 2014 annual mean (a) BrC-AAOD, and (b) contribution of BrC-AAOD to total AAOD at 440nm. Results are from the Modified_Age simulation.



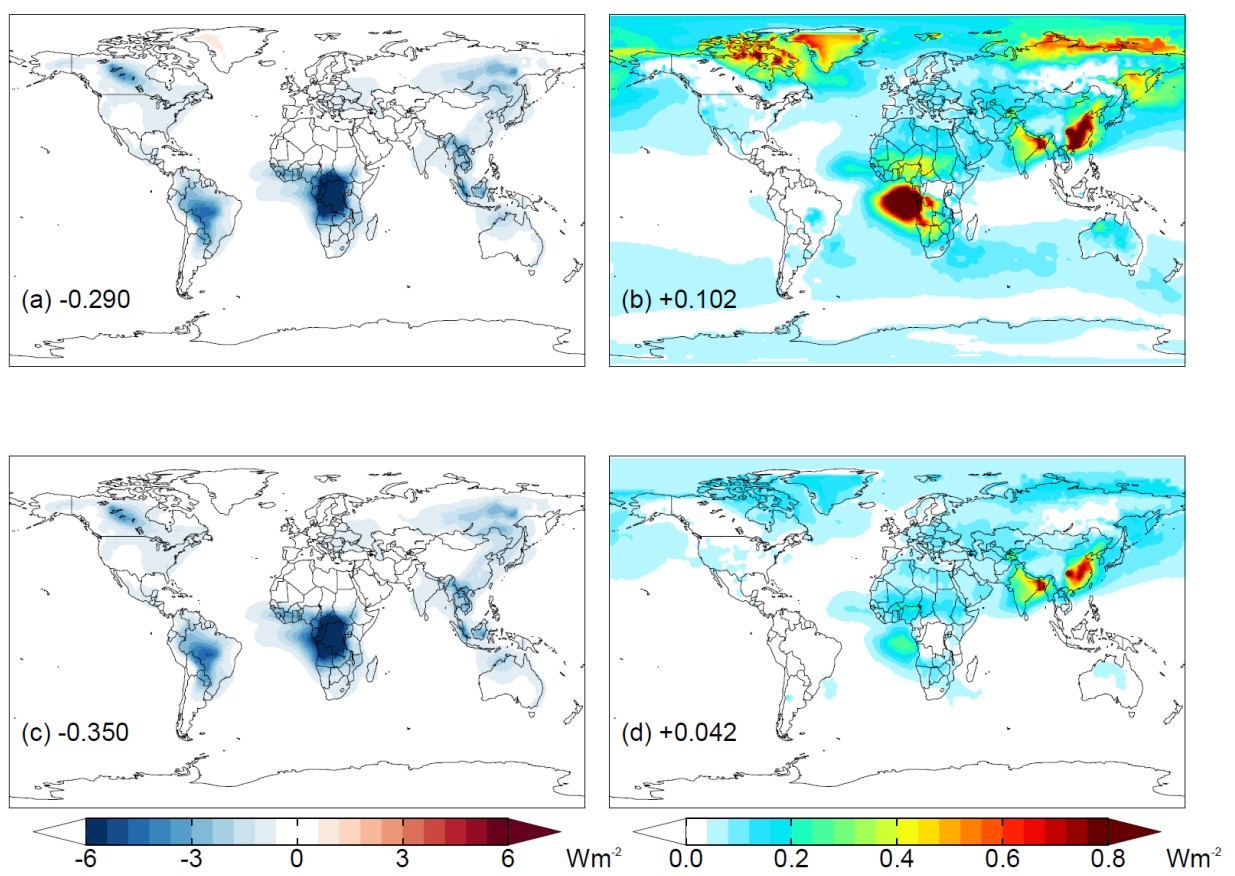

Figure 9. The global annual mean OA DRE (a and c) and BrC absorption DRE (b and d) at the top atmosphere (TOA) in 2014 from Base (a and b) and Modified_Age (c and d) simulations. Numbers indicate the global mean value in Wm⁻².