# Peer review of "Exploring the observational constraints on the simulation of brown carbon"

_Atmospheric Chemistry and Physics, 2017_

## Referee Comment (RC1) · Anonymous Referee #1 · 16 Aug 2017

Brown carbon (BrC) represents an emerging category of particulate organic compounds that can absorb solar radiation effectively in the spectral range of UV light. Although BrC is increasingly evolved in climate models, its (direct) radiative forcing remains highly uncertain, partly due to underestimation of organic aerosol (OA) mass by chemical transport models and lack of knowledge on optical properties of both primarily emitted and secondarily formed organic aerosols. The manuscript by Wang et al. investigated the observational constraints on the simulation of BrC by GEOS-Chem. The authors found that their modelling results on BrC absorption could be improved through increasing the OA mass associated with biomass burning (BB_OA), decreasing the mass absorption efficiency of BB_OA, and adding an aging scheme for BB_OA. The topic of this manuscript falls well within the scope of ACP. Although the manuscript

does not provide much scientific insights into the discrepancy between simulated and observed OA mass, the idea of involving a photo-chemical scheme in chemical transport models to simulate degradation of BrC is interesting. It could be accepted for publication given the authors could address the following concerns.

1. Page 1, line 19-20. An exaggerated statement was made here. Comparison of simulated and observed BrC has been performed by previous studies (e.g., Atmos. Chem. Phys., 16, 3413–3432, 2016). The statement in line 10-11, page 4 is more proper.

2. Page 3, line 17-19. These two sentences are questionable since BrC existing as coating materials on BC cores also absorb light. In the case of BrC coating, although the lensing effect is reduced relative to clear coating, this reduction can be overwhelmed by the effect of BrC shell absorption (e.g., Science of the Total Environment, 599–600, 1047–1055, 2017).

3. Page 3, the last paragraph. There have been many ground-based studies in which BrC absorption was directly measured by approaches similar to those used in DC3 and SEAC4RS. However, these studies were not mentioned and were not used to constrain simulation results on BrC. Although these studies could not provide information on BrC vertical distribution, surface BrC absorption could still be useful for the evaluation of simulated BrC.

4. Page 5, line 26. Suggest adding "with an OA/OC range of " before "1.34-1.65".

5. Page 7, equation 3. What is the difference between w and the absorption Ångström exponent (AAE)? According to equation 5, they should be the same. Please avoid using different terminologies for the same parameter.

6. Page 7, line 10-11. This point, i.e., biomass burning refers to open burning and does not include biofuel, should be clarified when biomass burning/biofuel were used for the first time in section 3.

7. Page 8, line 20. Why are the observed mass concentrations for externally mixed OA alone? HR-TOF-AMS cannot measure internally and externally mixed OA separately.

8. Page 10, the first paragraph. I am confused about sources of "the 80% bias". According to the discussions in line 6-23, it seems that this bias could be primarily attributed to neither the underestimate of POA emission factor nor the underestimate of biomass burning SOA. But then the authors said that this bias was due to "the underestimate of either the POA emission factor or biomass burning SOA" (line 26-27). These descriptions need to be revised to keep consistent. To my understanding, this bias indicated that the POA emission factor was considerably underestimated for biomass burning, although the actual BC/OA emission ratio would not be as low as 0.027 for biomass burning. In addition, please check "BC/OA emission factor" used in this paragraph. I think all of them should be "BC/OA emission ratio".

9. Page 13, line 24-25; and page 14, line 7. The authors are required to compare the AAE evolved here (which can be readily derived from the MAC values at different wavelengths) with those directly measured (e.g., Atmos. Chem. Phys., 15, 7841–7858, 2015), to see whether the assumed absorption wavelength dependence was reasonable. .

10. Page 13, line 25-26. The statement that "we assume that biomass burning SOA is equally absorbing as primary OA from biomass burning" does not agree with the description in section 3.2 that "For SOA, we assume that only aromatic SOA absorbs light", unless the authors assumed that all of the biomass burning SOA were from aromatic precursors.

11. Page 15, line 7 and elsewhere in the manuscript. I don't think the model assumptions applied in "Modified_Age" and "Modified_Simple" were really constrained from US fires observed during SEAC4RS. In fact, SEAC4RS was only used to evaluate or validate these assumptions.

12. Caption of Figure 7. Wavelength should be clarified.

---

## Referee Comment (RC2) · Anonymous Referee #2 · 18 Aug 2017

General comments

The paper of "Exploring the observational constraints on the simulation of brown Carbon" investigates the optical properties and DRE of BrC using GEOS-Chem model coupled with RRTMG model. They applied a photochemical scheme in the model to address the aging effect of BrC absorption and tested it against BrC absorption measurements from two aircraft campaigns. This study aims to "explore how assumptions for BrC sources, processing, and properties impacts the comparisons with these observational constraints and estimate the resulting global direct radiative effect of BrC under these conditions". While the authors addresses the topics listed in the paper, it is not immediately clear how significant the results actually are.

First, they need more constraints from observations near sources in addition to the

aircraft campaigns used in the study to test the photochemically whitening process for BrC absorption. Detailed comparison between Modified_Age and Modified_Simple should be provided to show the necessity and advantage of this aging scheme.

Second, the authors argued "DRE of BrC has been overestimated previously due to the lack of observational constraints from direct measurements and omission of the effects of photochemical whitening". However, they ignored some studies, which do not include this aging effect but show low DRE of BrC. For example, Hammer et at. (2016) estimated DRE of 0.03 W m2 for BrC constrained by OMI UVAI values, which is even lower than the result in this work (doi:10.5194/acp-16-2507-2016). Comparison with such studies may help to understand the factors contributing to the uncertainties in BrC absorption and to verify this aging scheme.

Finally, the MAC and the subsequently whitening process are strongly affected by the fraction of BrC associated with biomass burning. The authors assumed that the optical properties for biomass burning SOA are the same as those for biomass burning POA. But such assumption contradicts with their earlier statement that SOA is not absorber, at least not a significant absorber. Thus they may overestimate the fraction of BrC and underestimate the MAC for biomass buring OA.

In summary, this paper is well written and is easy to follow along. Its topic fits ACP and it is worthy of publication in ACP subject to addressing these and specific comments below.

Specific Comments

p. 1, line 23-24, the AAE is not constrained from absorption measurement

p. 2, line 10-11, as stated above, there are also studies with low DRE of BrC

p.4, line 28-30, the factor converting extract absorption to aerosol absorption is a function of aerosol size distribution. Is the factor of 2 consistent with the model assumption of OA size distribution in this study?

p.7, line 15, what is the density of OA used in the model? Will the assumption of the GMD of OA strongly affect its MAC?

p.8, line 16, from biomass burning and biofuel

p.9, line 24-25, high CH3CN and high CH3CN-OA may be due to the transport of plumes mixed with biomass burning and other sources. More evidence (e.g. enhancement ratio CH3CN/CO) is needed to support the conclusion of little contribution from sources other than biomass burning.

p. 10, line 2-3, the difference between 145% and 36% is $\sim$ 110%, not 80%

p.10, line 7-8, although lower than 0.03, the BC/OA of 0.027 should be still within the uncertainty range of biomass burning emission ratios

p. 10, line 29-30, only biomass burning OA is increased in FixBB. This won't affect the estimation of BrC absorption, but overestimates its contribution to total AAOD (the analysis in Sec. 5) as BC mass is still underestimated.

p. 14, line 17-18, the peak in the middle troposphere from SEAC4RS is not reproduced

p. 15, line 25, any explanation for high BrC absorption contribution in NA and Russia? The discrepancy between model and the observation is large as seen from the figure.

---

## Author Comment (AC1) · 23 Oct 2017

We thank the reviewer for his/her time and comments. We have made efforts to improve the manuscript accordingly, please find response for corresponding points below.

**Reviewer #1**

**Brown carbon (BrC) represents an emerging category of particulate organic compounds that can absorb solar radiation effectively in the spectral range of UV light. Although BrC is increasingly evolved in climate models, its (direct) radiative forcing remains highly uncertain, partly due to underestimation of organic aerosol (OA) mass by chemical transport models and lack of knowledge on optical properties of both primarily emitted and secondarily formed organic aerosols. The manuscript by Wang et al. investigated the observational constraints on the simulation of BrC by GEOS-Chem. The authors found that their modelling results on BrC absorption could be improved through increasing the OA mass associated with biomass burning (BB_OA), decreasing the mass absorption efficiency of BB_OA, and adding an aging scheme for BB_OA. The topic of this manuscript falls well within the scope of ACP. Although the manuscript does not provide much scientific insights into the discrepancy between simulated and observed OA mass, the idea of involving a photo-chemical scheme in chemical transport models to simulate degradation of BrC is interesting. It could be accepted for publication given the authors could address the following concerns.**

**1. Page 1, line 19-20. An exaggerated statement was made here. Comparison of simulated and observed BrC has been performed by previous studies (e.g., Atmos. Chem. Phys., 16, 3413–3432, 2016). The statement in line 10-11, page 4 is more proper.**

Our statement "this is the first study to compare simulated BrC absorption with direct ambient measurements" is correct given that previous studies have only had access to, and compared models with, indirect measurements. However, we have added text similar to that on page 4 to specifically indicate that we are referring to aircraft measurements here.

**2. Page 3, line 17-19. These two sentences are questionable since BrC existing as coating materials on BC cores also absorb light. In the case of BrC coating, although the lensing effect is reduced relative to clear coating, this reduction can be overwhelmed by the effect of BrC shell absorption (e.g., Science of the Total Environment, 599–600, 1047–1055, 2017).**

This is an interesting point. Given the limited measurements, it is not clear whether BrC coatings on BC cores also absorb light. We have added several sentences to discuss this point (page 3, line 22-24).

**3. Page 3, the last paragraph. There have been many ground-based studies in which BrC absorption was directly measured by approaches similar to those used in DC3 and SEAC4RS.**

**However, these studies were not mentioned and were not used to constrain simulation results on BrC. Although these studies could not provide information on BrC vertical distribution, surface BrC absorption could still be useful for the evaluation of simulated BrC.**

We thank the reviewer for this suggestion. However, previous direct observations of surface BrC absorptions are not particularly helpful in constraining model simulations as nearly all of these studies report data at low temporal resolution (e.g. 1-week or 1-month averages) without co-measured species. This makes it difficult to identify the sources of BrC and evaluate the model.

**4. Page 5, line 26. Suggest adding "with an OA/OC range of " before "1.34-1.65".**

Added.

**5. Page 7, equation 3. What is the difference between w and the absorption Ångström exponent (AAE)? According to equation 5, they should be the same. Please avoid using different terminologies for the same parameter.**

They are different. *w* refers to the wavelength dependence of imaginary part of refractive index (*k*); AAE refers to the wavelength dependence of absorption. These two kinds of wavelength dependence are different, since aerosol absorption is not linearly related to *k*.

**6. Page 7, line 10-11. This point, i.e., biomass burning refers to open burning and does not include biofuel, should be clarified when biomass burning/biofuel were used for the first time in section 3.**

We move this clarification to Section 3 (page 6, line 3).

**7. Page 8, line 20. Why are the observed mass concentrations for externally mixed OA alone? HR-TOF-AMS cannot measure internally and externally mixed OA separately.**

We have removed this sentence.

**8. Page 10, the first paragraph. I am confused about sources of "the 80% bias". According to the discussions in line 6-23, it seems that this bias could be primarily attributed to neither the underestimate of POA emission factor nor the underestimate of biomass burning SOA. But then the authors said that this bias was due to "the underestimate of either the POA**

emission factor or biomass burning SOA" (line 26-27). These descriptions need to be revised to keep consistent. To my understanding, this bias indicated that the POA emission factor was considerably underestimated for biomass burning, although the actual BC/OA emission ratio would not be as low as 0.027 for biomass burning. In addition, please check "BC/OA emission factor" used in this paragraph. I think all of them should be "BC/OA emission ratio".

In our discussion on page 10, we conclude that the 80% bias could not be attributed to the underestimate of POA emission factor only nor the underestimate of biomass burning SOA only. This means the bias is likely due to the integrated influence of these 2 factors. We agree with the reviewer that "BC/OA emission ratio" is a better than "BC/OA emission factor" in this paragraph and change the phrasing throughout.

**9. Page 13, line 24-25; and page 14, line 7. The authors are required to compare the AAE evolved here (which can be readily derived from the MAC values at different wavelengths) with those directly measured (e.g., Atmos. Chem. Phys., 15, 7841–7858, 2015), to see whether the assumed absorption wavelength dependence was reasonable.**

Thank you for the suggestion. We add a short discussion at the end of this paragraph (page 13, line 28-31) to address this point.

**10. Page 13, line 25-26. The statement that "we assume that biomass burning SOA is equally absorbing as primary OA from biomass burning" does not agree with the description in section 3.2 that "For SOA, we assume that only aromatic SOA absorbs light", unless the authors assumed that all of the biomass burning SOA were from aromatic precursors.**

We agree that this was unclear and now clarify this on page 7, line 26. The statement of SOA in Section 3.2 does not include biomass burning SOA.

**11. Page 15, line 7 and elsewhere in the manuscript. I don't think the model assumptions applied in "Modified_Age" and "Modified_Simple" were really constrained from US fires observed during SEAC4RS. In fact, SEAC4RS was only used to evaluate or validate these assumptions.**

The reviewer is correct. We clarify this point throughout the manuscript.

**12. Caption of Figure 7. Wavelength should be clarified.**

Added.

---

## Author Comment (AC2) · 23 Oct 2017

We thank the reviewer for his/her time and comments. We have made efforts to improve the manuscript accordingly, please find response for corresponding points below.

**Reviewer #2**

**General comments**

**The paper of "Exploring the observational constraints on the simulation of brown Carbon" investigates the optical properties and DRE of BrC using GEOS-Chem model coupled with RRTMG model. They applied a photochemical scheme in the model to address the aging effect of BrC absorption and tested it against BrC absorption measurements from two aircraft campaigns. This study aims to "explore how assumptions for BrC sources, processing, and properties impacts the comparisons with these observational constraints and estimate the resulting global direct radiative effect of BrC under these conditions". While the authors addresses the topics listed in the paper, it is not immediately clear how significant the results actually are.**

**First, they need more constraints from observations near sources in addition to the aircraft campaigns used in the study to test the photochemically whitening process for BrC absorption. Detailed comparison between Modified_Age and Modified_Simple should be provided to show the necessity and advantage of this aging scheme.**

We agree with the reviewer that it would be helpful to have observational constraints near sources to further evaluate the whitening process. However, unfortunately, to date, there are no such appropriate measurements. Previous direct observations of BrC absorption at the surface have low temporal resolution and have not been accompanied by measurements of other species (e.g. CO, NO, NO$_x$, etc.) needed to identify the photochemical aging state or transport time. We hope that future measurements (including perhaps during the upcoming FIREX and FireChem campaigns) will enable further evaluation of these schemes.

In Section 4.1, we show that the aging scheme improves the model simulation during the DC3 campaign. Both the Modified_Age and Modified_Simple schemes reflect the influence of aging on absorption, so a comparison between these two would not show the advantage of aging in capturing observations. In Section 5.2 we compare the simulated DRE between Modified_Age and Modified_Simple and find very small difference.

**Second, the authors argued "DRE of BrC has been overestimated previously due to the lack of observational constraints from direct measurements and omission of the effects of photochemical whitening". However, they ignored some studies, which do not include this aging effect but show low DRE of BrC. For example, Hammer et at. (2016) estimated DRE of 0.03 W m2 for BrC constrained by OMI UVAI values, which is even lower than the result**

**in this work (doi:10.5194/acp-16-2507-2016). Comparison with such studies may help to understand the factors contributing to the uncertainties in BrC absorption and to verify this aging scheme.**

Thank you for raising this point. We have added a discussion of Hammer et al. 2016 to the end of Section 5. However we emphasize that this study used indirect measurements to constrain the BrC DRE, and uncertainties on these measurements are challenging to estimate.

**Finally, the MAC and the subsequently whitening process are strongly affected by the fraction of BrC associated with biomass burning. The authors assumed that the optical properties for biomass burning SOA are the same as those for biomass burning POA. But such assumption contradicts with their earlier statement that SOA is not absorber, at least not a significant absorber. Thus they may overestimate the fraction of BrC and underestimate the MAC for biomass buring OA.**

We agree with the reviewer that the text was unclear on this point. We now clarify this on page 7, line 27. The statement regarding SOA absorption in Section 3.2 was not meant to include biomass burning SOA.

**In summary, this paper is well written and is easy to follow along. Its topic fits ACP and it is worthy of publication in ACP subject to addressing these and specific comments below.**

**Specific Comments**

**p. 1, line 23-24, the AAE is not constrained from absorption measurement**

The reviewer is correct. We have removed the sentence about AAE.

**p. 2, line 10-11, as stated above, there are also studies with low DRE of BrC**

We extend the DRE range here to represent all previous studies.

**p.4, line 28-30, the factor converting extract absorption to aerosol absorption is a function of aerosol size distribution. Is the factor of 2 consistent with the model assumption of OA size distribution in this study?**

The factor of 2 is related to the size distribution of BrC field measurements at 3 sites (Liu et al., 2013). In these measurements, the mass mean diameter (MMD) of OA is 500nm with standard

deviation (δ) of 1.5 – 2.4. With a standard deviation of 1.8, this MMD can be transferred to a count mean diameter (CMD) of ~180nm. This size distribution is very close to our assumption (CMD = 180nm, δ=1.6). We add a sentence to clarify this point in Section 2, page 4, line 20-31.

**p.7, line 15, what is the density of OA used in the model? Will the assumption of the GMD of OA strongly affect its MAC?**

The density of OA is assumed to be 1.3 $g/m^3$. We add this to the text in Section 3.2, page 7, line 13.

The GMD of OA could affect its MAC strongly but this influence is non-linear. For example, with δ=1.6 and refractive index of BB BrC, a 50% decrease in GMD causes 1% difference in MAC, however, a 50% increase in GMD will increase MAC by 35%. As replied in last comment, we use the same GMD value as measured BrC in filed observations.

**p.8, line 16, from biomass burning and biofuel**

Changed (typo).

**p.9, line 24-25, high CH3CN and high CH3CN-OA may be due to the transport of plumes mixed with biomass burning and other sources. More evidence (e.g. enhancement ratio CH3CN/CO) is needed to support the conclusion of little contribution from sources other than biomass burning.**

There is no significant CH3CN/CO enhancement during the identified BP (the ratio during BP is only ~15% higher than the average during the campaign). However, we find not only high CH3CN and high CH3CN-OA correlation but also high OA-BC correlation, high BC and OA concentrations during BP. We think this is enough to identify the biomass burning influenced periods. We have clarified this point in the Section 4.1.

**p. 10, line 2-3, the difference between 145% and 36% is _ 110%, not 80%**

The difference between 145% and 36% in this context is 80%. After a 36% increase, we still need ~80% increase to get the 145% difference: $(1+0.36) \times 1.8 - 1 = 1.45$.

**p.10, line 7-8, although lower than 0.03, the BC/OA of 0.027 should be still within the uncertainty range of biomass burning emission ratios**

We agree that this value should be within the uncertainty range of BC/OA emission ratio, but it seems unlikely that all fire emissions fall below the range of emission ratio (0.03-0.06) for biomass burning as given in GFED. Our statement does not preclude this possibility, we simply indicate that it is unlikely.

**p. 10, line 29-30, only biomass burning OA is increased in FixBB. This won't affect the estimation of BrC absorption, but overestimates its contribution to total AAOD (the analysis in Sec. 5) as BC mass is still underestimated.**

In our analysis of DC3 data, we adjust both the OA and BC biomass burning mass concentrations upwards to match the observations (as indicated in the previous sentence). A further increase to OA is applied to correct for an underestimate in the OA emission factors and/or missing biomass burning related SOA. Therefore biases in biomass burning are addressed for BC and OA and correctly reflected in the AAOD calculations.

**p. 14, line 17-18, the peak in the middle troposphere from SEAC4RS is not reproduced**

The sentence "… during DC3 and SEAC4RS except for altitudes above 10 km." is changed to "… during DC3 and SEAC4RS at the altitudes with enough data points below 10 km."

**p. 15, line 25, any explanation for high BrC absorption contribution in NA and Russia? The discrepancy between model and the observation is large as seen from the figure.**

This is an interesting point. The absorption properties and/or BrC contribution to total OA may be different for different fires/biofuel combustion sources. Our model assumptions constrained from the US fires are able to capture the observed BrC absorption contributions in Europe, but have large discrepancy in Alaska and Russia. This suggests that the BrC absorption properties and/or BrC contribution to total OA could be very different between Alaska/Russia and US/Europe. Due to the limitation of measurement data, we are not able to undertake any further analysis of this discrepancy, but we add text acknowledging these discrepancies in page 15, line 17-18.

---

## Author Response (AR2)

We thank the reviewers for their re-review of this manuscript and their suggestions.

**Response to reviewer:**

**The authors have not fully addressed the first comments from reviewer #2: "First, they need more constraints from observations near sources in addition to the aircraft campaigns used in the study to test the photochemically whitening process for BrC absorption. Detailed comparison between Modified_Age and Modified_Simple should be provided to show the necessity and advantage of this aging scheme". As it was pointed out earlier, the aging scheme should be evaluate both near sources and along the transport (e.g. outflow regions). Although the authors argued that this evaluation couldn't not be done because of "no such appropriate measurements", such discussion/claim should be included in the text.**

We have now added the discussion of the lack of appropriate measurements in Section 6, page 19 line 5 – 9.

**In addition, the authors' reply that "we compare the simulated DRE between Modified_Age and Modified_Simple and find very small difference" can not prove the "necessity and advantage of this aging scheme". In contrast, this may simply indicate that the MAC for biomass burning SOA is overestimated before, and there is no need to apply this whitening process at all.**

Although there is small difference in global DRE between Modified_Age and Modified_Simple, there are differences in the spatial distribution of BrC absorption. For example, the ratio of mean absorption DRE over land to ocean is 1.52 for the Modified_Age scheme, 10% larger than for the Modified_Simple scheme (1.38). This is because whitening reduces BrC absorption in remote oceans than in near-source land regions. Therefore, the whitening scheme could be important for simulating the regional distribution of BrC absorption. Unfortunately, we do not yet have the necessary suite of observations to test these differences. We have added text to this effect to Section 5.1 of the manuscript.

As the reviewer suggests, since both the whitening process and a lower MAC result in a lower simulated BrC absorption, one could argue that it is the overestimate of BB-SOA MAC, instead of the missing whitening process, that is the source of the model bias. However, if we assumed BB-SOA MAC=0 (and leave the POA MAC unchanged), our simulation would still overestimate the observed absorption. Therefore, whitening (which has been observed in the field) is the only means by which we can reconcile model and observations. We have clarified this in Section 4.1

**Finally, please give reference for the statement that "OA from fresh BB emissions exhibits similar absorption properties as observed in laboratory studies" in p 12. line 5-6.**

We apologize if this was unclear – the statement follows the logic of the earlier context. The initial MAC is based on the laboratory study of Saleh et al. (2014) (Section 3.2) and is around the median value of previous experiments (reference now added in the text).

[revised manuscript text omitted]